Does the sex difference in competitiveness decrease in selective sub-populations? A test with intercollegiate distance runners

Deaner Robert O. 1 robert.deaner@gmail.com
Lowen Aaron 2
Rogers William 1
Saksa Eric 1
1 Department of Psychology, Grand Valley State University , United States
2 Department of Economics, Grand Valley State University , United States
Badcock Nicholas
Electronic publication date: 2015 Apr 21
Publication date: 2015
Volume: 3
Electronic Location ID: e884
Received 2015 Jan 28; Accepted 2015 Mar 19
Copyright: © 2015 Deaner et al.
Copyright year: 2015
Copyright holder: Deaner et al.
License: This is an open access article distributed under the terms of the Creative Commons Attribution License, which permits unrestricted use, distribution, reproduction and adaptation in any medium and for any purpose provided that it is properly attributed. For attribution, the original author(s), title, publication source (PeerJ) and either DOI or URL of the article must be cited.
License URL: https://creativecommons.org/licenses/by/4.0/

Keywords: Gender differences, Competition, Sports, Evolutionary psychology, Distance running, Title IX, Athletics, Motivation, Elite athletes, Preferences

Funding: This research was supported by the Department of Psychology at Grand Valley State University. The funders had no role in study design, data collection and analysis, decision to publish, or preparation of the manuscript.

==============================
Sex differences in some preferences and motivations are well established, but it is unclear whether they persist in selective sub-populations, such as expert financial decision makers, top scientists, or elite athletes. We addressed this issue by studying competitiveness in 1,147 varsity intercollegiate distance runners. As expected, across all runners, men reported greater competitiveness with two previously validated instruments, greater competitiveness on a new elite competitiveness scale, and greater training volume, a known correlate of competitiveness. Among faster runners, the sex difference decreased for one measure of competitiveness but did not decrease for the two other competitiveness measures or either measure of training volume. Across NCAA athletic divisions (DI, DII, DIII), the sex difference did not decrease for any competitiveness or training measure. Further analyses showed that these sex differences could not be attributed to women suffering more injuries or facing greater childcare responsibilities. However, women did report greater commitment than men to their academic studies, suggesting a sex difference in priorities. Therefore, policies aiming to provide men and women with equal opportunities to flourish should acknowledge that sex differences in some kinds of preferences and motivation may persist even in selective sub-populations.

Introduction

Sex differences in some preferences and motivations are well established. Notably, compared to women, men generally show greater desire for sex in short-term or uncommitted contexts (Baumeister, Catanese & Vohs, 2001; Schmitt, 2005), greater preference for occupations that involve working with things rather than with people (Lippa, 1998; Konrad et al., 2000; Su, Rounds & Armstrong, 2009), greater motivation to prioritize the professional sphere over the domestic one (Browne, 2002; Hakim, 2006), and, in several domains, greater willingness to take risks (Wilson & Daly, 1985; Byrnes, Miller & Schafer, 1999; Croson & Gneezy, 2009).

One outstanding question is whether sex differences in preferences and motivations persist when men and women are drawn from highly selective sub-populations or are otherwise equated on domain-relevant traits. For example, in a recent review of women’s lesser representation in science, Ceci, Ginther, Kahn, and Williams (Ceci et al., 2014) suggested that policy makers should take seriously the possibility that women’s lesser representation is partly due to the fact men and women of similar scientific ability may differ considerably in various preferences and motivations.

Unfortunately, there is little empirical data bearing directly on this question. One study explored the lifestyle preferences of adults in their mid-30s who had been identified as profoundly gifted as children or as top math or science graduate students ten years earlier. In these selective sub-populations, men were more likely than women to prioritize their careers over their families, although the difference was smaller among those with no children (Ferriman, Lubinski & Benbow, 2009). There have also been relevant studies of financial decision making (Croson & Gneezy, 2009). These studies typically report that the sex difference in risk taking substantially weakens (Dwyer, Gilkeson & List, 2002; Halko, Kaustia & Alanko, 2012) or disappears (Johnson & Powell, 1994; Atkinson, Baird & Frye, 2003) when financial knowledge or wealth are controlled.

Competitiveness

We suggest that progress in addressing this issue may come from investigating the preference for engaging in direct competition (hereafter “competitiveness”).1 Competitiveness is an appropriate focus because it constitutes a core component of sex differences, and the sex difference in competitiveness is thought to have practical implications (Wilson & Daly, 1985; Croson & Gneezy, 2009; Niederle & Vesterlund, 2011; Benenson, 2013). Moreover, evidence for a sex difference has emerged from many areas: on surveys, men report greater enjoyment of competition in general (Houston et al., 2005; Piko et al., 2010), greater desire to win in interpersonal situations (Spence & Helmreich, 1983; Gill, 1988), and greater desire to strive for success relative to others in sports (Gill, 1988; Merten, 2008; Findlay & Bowker, 2009); in laboratory games, men are more likely than women to choose competitive rather than non-competitive compensation schemes (Croson & Gneezy, 2009; Niederle & Vesterlund, 2011); in their free time, boys and men play video games more often than girls and women, particularly competitive games, and males are more likely to report that competition motivates them to play (Lucas & Sherry, 2004; Hartmann & Klimmt, 2006); and, finally, boys and men participate in sports substantially more often than girls and women, although there is no consistent difference for non-competitive physical activity or exercise (Stamatakis & Chaudhury, 2008; Lunn, 2010; Deaner et al., 2012).

To our knowledge, only one previous study tested whether the sex difference in competitiveness decreases in selective sub-populations. Houston, Carter, and Smither (Houston, Carter & Smither, 1997) reported that among active professional tennis players (n = 130), there was no sex difference in the competitiveness scale of the Sport Orientation Questionnaire (SOQ: Gill & Deeter, 1988) and that women scored significantly higher (Cohen’s d = − 0.5) on the Competitiveness Index (Smither & Houston, 1992). These results support the hypothesis that the sex difference in competitiveness disappears in more selective sub-populations. Nonetheless, tennis might be an unusual sport because this study also found no sex difference with either competitiveness measure among amateur tennis players (n = 92).

Other studies, although not specifically designed to address the question of interest, have yielded relevant data. Gill & Dzewaltowski (1988) investigated the competitiveness of undergraduate students and intercollegiate student-athletes (NCAA Division I) with the SOQ. Athletes reported significantly greater competitiveness than non-athletes (d = 1.5), and men reported significantly greater competitiveness than women (d = 0.5). Effect sizes could not be calculated for sex difference within each sample, but the authors reported there was no significant interaction between sex and population. These findings support the hypothesis that the sex difference in competitiveness is not attenuated in more selective sub-populations. Nevertheless, this conclusion must be viewed cautiously because the athletes were drawn from several sports, competitiveness varied significantly across sports, and the authors did not control for this possible confound. Moreover, the study’s modest sample size (n = 213) suggests it may have lacked sufficient power to detect the interaction of interest. Three other studies reported significantly greater male competitiveness with the SOQ in selective athletic populations (Hellandsig, 1998: “talented” teens in Norway, n = 175, d = 0.5; (Jamshidi et al., 2011): “elite” athletes in Iran, n = 688, d = 1.0; (Poiss et al., 2004): NCAA Division III student-athletes, n = 304, d = 0.4). However, none of these studies addressed the potentially confounding role of variation across sports or included a less selective comparison group.

Therefore, it remains unresolved whether the sex difference in competitiveness disappears or becomes attenuated in selective athletic sub-populations. Here, we address this question for intercollegiate distance runners.

Distance running as a test domain

Distance running is an excellent domain for assessing variation in competitiveness because the motivation to run varies substantially among runners. While some runners are primarily motivated by competition, most mainly run for other reasons, such as affiliation, health, and life meaning (Masters, Ogles & Jolton, 1993; Ogles & Masters, 2003). Distance running is also advantageous because, unlike many sports, it is generally accessible, acceptable, and popular for both men and women, and the financial incentives do not appear to favor men (Deaner, 2013).

There is mounting evidence for a sex difference in competitiveness in distance running. First, more male than female runners report that competition motivates them to run (Callen, 1983; Johnsgard, 1985; Ogles & Masters, 2003). Second, male runners are more likely to choose to participate in competitive contexts. Among masters runners (age 40+) in the U.S., male and female participation in road races is equivalent, yet at track meets, where runners are twenty times more likely to run fast (relative to age-specific, sex-specific standards), men participate about three times as often as women (Deaner, Addona & Mead, 2014). Similarly, a study reported that when they have the option of entering a single-sex competitive road race or a single-sex non-competitive road race held in the same location on the same day, men were substantially more likely than women to select the competitive race (Garratt, Weinberger & Johnson, 2013). Third, at U.S. road races, roughly three times as many men as women run fast relative to sex-specific world class standards, and the best supported explanation for this pattern is that more men engage in the training necessary for faster performances (Deaner, 2006b; Deaner, 2013; Deaner & Mitchell, 2011). Finally, there is a robust sex difference in pacing in the marathon: men are three times more likely than women to slow dramatically, and this likely reflects, in part, men’s greater risk taking, a component of competitiveness (March et al., 2011; Deaner et al., 2015).

Studying intercollegiate distance runners offers several advantages for addressing the question of whether the sex difference in competitiveness disappears or becomes attenuated in selective sub-populations. One advantage is that sub-population selectivity can be addressed by grouping runners according to their running ability. We will measure ability using runners’ best timed race performances and also by their collegiate team’s National Collegiate Athletic Association athletic division (i.e., NCAA Division I, Division II, or Division III). These measures have complementary strengths. Best timed race performance is a direct ability measure but, for various reasons, not all runners have the opportunity to perform (or perform to the best of their ability) at commonly contested distances. Athletic division, by contrast, moderately correlates with timed performance (see Results) and is readily reported by all runners. Division I runners are generally fastest, Division II runners are intermediate, and Division III runners are slowest.

A second advantage of studying intercollegiate distance runners is that competitiveness can be assessed with multiple measures. Specifically, we will use a general sports competitiveness instrument, the SOQ (Gill & Deeter, 1988), a motivational instrument designed specifically for distance runners (Motivation for Marathoners Scales; MOMS: Masters, Ogles & Jolton, 1993), and new items addressing motivation to compete and train at a selective or elite level after college. In addition, we will use training volume as another indicator of competitiveness (Deaner, 2013). Training volume, assessed as distance run per week or running sessions per week, is consistently associated with better distance running performance (Bale, Rowell & Colley, 1985; Hagan et al., 1987; Deaner et al., 2011) and self-reported competitiveness (Masters, Ogles & Jolton, 1993; Ogles & Masters, 2003; Deaner et al., 2011).

A third advantage of studying intercollegiate distance runners is that explanations for the possible sex difference in competitiveness can be explored. In particular, we will test whether sex differences in competitiveness in running can be attributed to sex differences in injuries or obligations outside of sports, including childcare.

The purpose of this study, therefore, is to test, within the domain of distance running, whether the expected sex difference in competitiveness decreases in increasingly selective sub-populations. Its strengths are that it uses two measures of selectivity, several measures of competitiveness and investigates possible causes of sex differences.

Methods

Research approval

All human subjects were treated in accordance with established ethical standards. The Chair of the Human Research Review Committee at Grand Valley State University reviewed the study protocol [427545-1] and certified it as approved and exempt from full committee review on February 28, 2013.

Recruitment

To identify colleges and universities for possible recruitment, we made use of the National Collegiate Athletic Association’s (NCAA) online directory of sports sponsorship (http://perma.cc/8BLQ-57B3). In February 2013, we created a list of all institutions (N = 976) with men’s varsity intercollegiate cross country teams participating in any of the NCAA’s three divisions, Division I, Division II, or Division III (hereafter, DI, DII, and DIII). With the exception of Jacksonville State, all of the institutions on this list sponsored both men’s and women’s cross country teams. In addition, there were approximately 80 NCAA institutions that sponsored women’s but not men’s cross country teams. These were mainly women’s colleges, and we did not include them on our recruitment list.

We selected, at random, half of the institutions on our recruitment list and then determined which of these allowed public access to student email directories. Out of 488 institutions, 130 (27%) allowed email access, 67 of 164 (41%) for DI, 21 of 127 (17%) for DII, and 42 of 197 (21%) for DIII. For these schools, we attempted to acquire the names of all student-athletes who were listed on the male or female cross-country team roster at each institution’s athletics website. We then attempted to acquire each student-athlete’s email. We were successful in 2,163 of 2,245 cases (92%) for men and 2,218 of 2,362 cases (94%) for women. Most failures were related to the runner having a name shared by others at their institution; we did not acquire an email unless we were confident it uniquely corresponded with a student-athlete listed on the roster.

We sent out recruitment emails on March 19, 2013. This email contained a request for participation in a survey study of motivation, training and performance in distance runners. It offered no incentives and stated that participation would take roughly 10 min.

Approximately 1,460 individuals of 4,381 (33%) that we contacted opened the survey link and consented to participate, although some merely skimmed the survey and answered few or no questions; 1,147 individuals (26% of those contacted) provided sufficient data for inclusion in at least some analyses. Besides those who elected not to open the survey link or who opened the survey but elected not to complete it, the response rate was diminished because some emails, roughly 1%, “bounced back” with a message indicating the accounts were no longer active. Moreover, an unknown number of emails may have been filtered by software and thus not read. Only responses completed prior to March 24, 2013 were included in the final data set.

Participants

Of the 1,147 participants, 608 were men (28.1% response rate) and 539 were women (24.3%). Participants’ mean age was 20.1 years, and all were at least 18 years of age. Participants reported an average of 2.2 previous seasons of collegiate cross country. They reported an average of 5.5 total previous seasons of collegiate distance running, i.e., summing the number of seasons of cross-country, number of seasons of indoor track and field, and number of seasons of outdoor track and field. Twenty-five percent were in their first year of undergraduate studies, 25% were second years, 23% were third years, 21% were fourth years, 3% were fifth years, and 3% were graduate students. Forty-six percent were DI athletes, 11% were DII athletes, and 43% were DIII athletes. Eighteen percent reported finishing among the top 10 individual cross-country performers in their conference meet at least once, and 5% reported finishing among the top 50 individual cross-country performers in their national meet at least once. There were no significant sex differences in any of these characteristics (all ps > .08).

Women reported being more likely to receive athletically-related financial aid (χ2(3, n = 1,145) = 12.5, p = .006). Eight percent of women received full athletic scholarships, 26% received partial athletic scholarships, 4% received “other financial aid related to athletic ability,” and 62% reported receiving no aid; the corresponding values for men were 4.3%, 22%, 6%, and 69%. Women also reported being somewhat more highly recruited than men (χ2(3, n = 1147) = 10.6, p = .01). Sixteen percent reported being highly recruited, 35% reported being moderately recruited, 28% reported being lightly recruited, and 20% reported not being recruited at all; the corresponding values for men were 15%, 31%, 37%, and 17%. These differences likely reflect that, across all DI and DII institutions, there is roughly 40% more athletically-related financial aid available for cross country and track and field for women than for men (NCAA Research, 2011). DIII institutions are not permitted to offer athletically-related financial aid.

Representativeness of the sample

To evaluate the representativeness our sample, we focused on runner’s abilities, specifically whether runners had ever finished among the top 50 individual cross-country performers in their division’s national meet. Running ability is an appropriate focus because it is known to be associated with competitiveness (Masters, Ogles & Jolton, 1993; Ogles & Masters, 2003; Deaner et al., 2011). Competitiveness is the study’s focal topic and thus its relation to response propensity warrants investigation (Montaquila et al., 2007).

We assessed the occurrence of achieving the top 50 benchmark in our sample by including it as a survey item. We assessed the occurrence of this benchmark in the sub-population of NCAA distance runners that we attempted to recruit (rather than the full population of all NCAA distance runners) by first gathering cross-country championship results for each of the three divisions for the years, 2009–2012.2 After eliminating those who finished in the top 50 more than once, we had 703 unique names, 350 men and 353 women. Of the 350 men who finished in the top 50 at least once, 62 were sent the survey, 4 did not have available emails, and the remainder were not on our list of potential recruits (see above, “Recruitment”). Of the 353 women who finished in the top 50 at least once, 41 were sent the survey, 8 did not have available emails, and the remainder were not on our list of potential recruits.

In our sample, 30 of the 603 male and 26 of the 532 female respondents (5% of each) answered “Yes” to the item about having been a top 50 performer. (Five men and seven women who participated in some parts of the survey did not answer this question.) If all respondents who answered this question did so honestly and accurately, then 48% (30 of 62) of the top male performers and 63% (26 of 41) of the top female performers who were recruited participated in our survey. This sex difference in the likelihood of participation was not significant (χ2(1, n = 103) = 2.2, p = .13). By contrast, 27% of non-top 50 males (573 of 2,096) and 24% of non-top 50 females (506 of the 2,170) participated in our survey, a sex difference which was significant (χ2(1, n = 4, 266) = 7.7, p = .006). For both men and women, top 50 finishers were significantly more likely to participate than those who never finished in the top 50 (men: χ2(1, n = 2, 158) = 13.3, p = .0003; women: χ2(1, n = 2, 211) = 34.5, p < .0001). Thus, faster runners were substantially more likely to participate than slower runners, and men were more likely than women to participate, although this sex difference was due to greater participation by slower male runners.

Survey

The survey was presented on SurveyMonkey, a commercial survey platform. Respondents first provided written consent to participate. Next were items addressing training, injuries, and performance, including the following items that were examined in this study: (1) “When training in college, what was the highest average running mileage per week you maintained for a 4 week period? If you didn’t keep track or can’t recall, please leave this blank.” (We inquired about miles, rather than kilometers, because most U.S. runners monitor their running distance in miles.); (2) “When training in college, what was the highest average number of running sessions per week you maintained for a 4 week period? If you didn’t keep track or can’t recall, please leave this blank.”; (3) “When running regularly in college, how many sessions of aerobic cross-training did you usually do per week? This might include elliptical, biking, swimming, aqua running or other aerobic exercise. If you didn’t keep track or can’t recall, please leave this blank.”; (4) “How many times in your college career did you miss at least 2 weeks of training because of illness or injury?” Options for this item were 0, 1–3, 4–6, 7–9, and 10+; (5) “Please list your best collegiate track performance for two of the following distances. If you competed at more than two distances, please provide information on the distances where you performed best”. Options were 800 m, 1,500 m, 1,600 m, 1,609 m (mile), 3,000 m, 3,200 m, 3,218 m (2 mile), and 5,000 m; (6) “Have you ever finished as one of the top 10 individual cross-country performers in your conference meet?” Options were yes and no; (7) “Have you ever finished as one of the top 50 individual cross-country performers in your division’s national meet?” Options were yes and no.

Next were seven items designed to desire address motivation to train and compete as a highly competitive or professional runner after college. Options for each item ranged from one (strongly disagree) to five (strongly agree). The items were: (1) “I plan to continue racing competitively after my collegiate career is over.”; (2) “I might try making it as a professional runner.”; (3) “Qualifying for the Olympic trials or a national team is a long-term goal of mine.”; (4) “I’d love to be a professional athlete.”; (5) “I would like to take my training to the next level.”; (6) “I look forward to training much less after my collegiate running career is over.” (reverse scored); and (7) “If I had the opportunity, I would train much more than I currently do.”

There were then four items addressing possible injuries and commitment to academics. Options for the first three items ranged from one (strongly disagree) to five (strongly agree). The items were: “I have often had to limit my training because of injuries and illness.”; “I am committed to doing the best that I can in school.”; and “I take my education at least as seriously as my running.” The fourth item was, “How many hours per week do you usually study?” Options were 0–5, 6–10, 11–15, 16–20, 21–25, 26+.

Next were several additional items concerning academics (e.g., grade point average, academic major), but these were not included in this study. Then there several items addressing social behavior, including the following items that were assessed below: (1) “I already have (or had) at least one child.” Options were yes and no; (2) “I am likely to have children in the next 3 years.” Options ranged from 1 (strongly disagree) to 5 (strongly agree). Following this were several demographic items, most of which were noted earlier in the Methods section.

The next component was the Motivations of Marathoners Scales (MOMS: Masters, Ogles & Jolton, 1993), which consists of 56 items that are rated as to the degree to which the runner considers them a reason for training and running in a marathon or distance race (Masters, Ogles & Jolton, 1993). Items represent nine internally consistent motivational constructs: affiliation (6 items), competition (4), health orientation (6), life meaning (7), personal goal achievement (6), psychological coping (9), recognition (6), self-esteem (8), and weight concern (4). Each item is rated on a one (not a reason) to seven (a very important reason) scale. The score for each scale is calculated by averaging the score for each item included in the scale. Evidence for the internal consistency (Cronbach’s alphas range from .80 to .93), test-retest reliability (rs range from .71 to .90), and factorial and construct validity of the scales has been presented previously (Masters, Ogles & Jolton, 1993; Masters & Ogles, 1995; Ogles, Masters & Richardson, 1995). Because the scales were developed in mainly recreational samples of distance runners, and runners in the current study were younger and generally much faster, we analyzed the factor structure of these 56 items. As detailed in the Supplemental Information, we found it similar to the original structure, and the structure was similar in men and women.

Participants then completed the Sport Orientation Questionnaire (SOQ), which consists of 25 items that are rated on a scale ranging from 1 (strongly disagree) to 5 (strongly agree) (Gill & Deeter, 1988). Items represent three internally consistent motivational constructs: competitiveness (13 items), win orientation (6), and goal orientation (6). The score for each scale is calculated by averaging the score for each item included in the scale. Evidence for the internal consistency (Cronbach’s alphas range from .79 to .95), test-retest reliability (rs range from .73 to .89), and factorial and construct validity of the scales has been presented previously (Gill & Deeter, 1988). These scales were developed in both general and highly athletic samples of high school and college students. Nonetheless, we explored their factor structure among the distance runners in the current study. As detailed in the Supplemental Information, we found it similar to the original structure, and the structure was similar in men and women.

Comparisons of men’s and women’s running ability and training

Cross-country courses vary substantially in terrain and accuracy of measured distances, so we focused on best track performances, which are far more standardized. However, not all runners have participated in the same track distances, and making comparisons across distances is difficult because it appears to be easier to run relatively fast (e.g., proportionally close to a word record) at shorter distances (Deaner, 2006b). Hence, we based our ability classifications on the race distance where runners most frequently reported a best performance, the 5,000 m run; 632 runners (299 women) reported a best performance for this distance.

Comparisons of men’s and women’s performance in the 5,000 m (and all other distances) are complicated by the fact that, for any given level of talent and training, men run roughly 10% faster than women (Sparling, O’Donnell & Snow, 1998; Thibault et al., 2010). This gap is believed to mainly reflect men’s greater maximal oxygen uptake, which is mediated by several physiological factors, including men’s greater hemoglobin concentration, lesser body fat, and greater muscle mass per unit of body weight (Sparling, O’Donnell & Snow, 1998; Thibault et al., 2010). It is difficult to make a fair correction for this difference (e.g., reducing women’s performance durations by 10%) because the sex difference in performance is roughly 10% among the fastest runners but is often observed to be 15%–20% among somewhat slower runners (Sparling, O’Donnell & Snow, 1998; Deaner, 2006b; Thibault et al., 2010). Therefore, we assigned runners into one of four sex-specific ability quartiles based on 5,000 m performance. For the 299 women who reported a performance, the cut-off to be included in quartile four, the fastest women’s group was 17:54 or faster; the cut-off to be in the slowest group was 19:30 or slower. For the 333 men who reported a performance, the corresponding performance cut-offs were 14:47 or faster and 15:58 or slower.3

Comparing men’s and women’s training volumes in term of distance is also complicated by the performance gap between male and female runners. This is because slower performers generally train more slowly and hence will accumulate less training distance for the same duration of running. Therefore, although we explored absolute mileage per week, most comparisons focused on mileage per week after increasing each women’s reported mileage by 22%. Below we refer to this as “adjusted mileage.” We chose a 22% adjustment because the sex difference in the inter-quartile grouping cut-offs ranged from 21%–22%. Moreover, NCAA top performance lists indicate a similar sex difference in each athletic division (http://perma.cc/WWQ7-NHKY). For example, for the 2013 Outdoor Championships, the sex difference for 500th best 5,000 m performance was 19% among DI runners, 23% among DII runners, and 22% among DIII runners. Although we know of no previous study that has adjusted women’s training distances, previous studies have shown that adjusting women’s running performances can prevent the over-estimation of sex differences (Deaner et al., 2011; Deaner et al., 2015).

Data inclusion

We only considered responses from individuals who reported both their sex and athletic division. Among these individuals, there were a substantial number who did not complete all items, particularly the MOMS and SOQ scales, which were administered later in the survey; we generally included data from such individuals. However, in completing multi-item scales, some participants answered some but not all items; we did not include these responses. This occurred for the 56-item MOMS (n = 151 excluded, 69 women), the 25-item SOQ (n = 51, 20 women), and the 7-item elite competitiveness scale (n = 4, 2 women). Because they seemed unlikely to represent sincere responses, we also excluded the SOQ and MOMS data from one man who reported “1” for all 71 items on these scales; we excluded data from one woman who reported “1” for all 25 items of the SOQ scale. All other individuals showed variability across these multi-item scales.

Some responses to the performance and training questions were not credible, presumably because runners had poor recall or did not understand the items (e.g., they provided monthly rather than weekly estimates of training volume). Therefore, to minimize their effects, we deleted the following highly improbable responses: 5,000 m performances faster than the current world record for one’s sex (n = 4, 2 women), regularly completing more than 24 running sessions per week (n = 41, 17 women), and running greater than 200 miles (322 km) per week for a 4 week period (n = 5, 3 women).

Statistics

We conducted analyses with Statistica 6.1 (Statsoft Inc., Tulsa, Oklahoma USA). Following the convention in sex differences research, a positive effect size (Cohen’s d) indicates a greater value for men; a negative effect size indicates a greater value for women. All statistical tests were two tailed and α was set at .05.

Results

Overall sex differences in competitiveness and training

Similar to previous research with the SOQ (Gill & Deeter, 1988; Poiss et al., 2004; see also Gill & Dzewaltowski, 1988; Hellandsig, 1998; Merten, 2008; Jamshidi et al., 2011), men reported significantly greater competiveness (d = 0.56) and win orientation than women (d = 0.48), but there was no significant sex difference for goal orientation (d = 0.07; Table 1). Consistent with previous research using the MOMS (Ogles, Masters & Richardson, 1995; Ogles & Masters, 2003; Deaner & Mitchell, 2011), men reported significantly greater endorsement than women for competition (d = 0.61) and, somewhat unexpectedly, also for goal achievement, although the effect size for this was modest (d = 0.21). For the other seven MOMS sub-scales, there was no significant sex difference or women’s scores were significantly higher than men’s (Table 1).

Table 1 Sex differences in motivation and training.

Topic or instrument	Sub-scale or item	Men	Women	t	Cohen’s d	
		M	SD	n	M	SD	n			
SOQ										
	Competitiveness	4.34	0.5	555	4.03	0.6	496	9.07***	0.56	
	Goal orientation	4.43	0.5	555	4.39	0.5	496	1.16	0.07	
	Win orientation	3.67	0.8	555	3.28	0.8	496	7.73***	0.48	
MOMS										
	Affiliation	4.16	1.4	514	4.65	1.4	461	5.50***	−0.35	
	Competition	5.43	1.2	514	4.61	1.5	461	9.50***	0.61	
	Goal achievement	6.17	0.8	514	5.99	0.9	461	3.31***	0.21	
	Health	4.25	1.4	514	4.71	1.2	461	5.36***	−0.35	
	Life meaning	3.97	1.5	514	4.40	1.5	461	4.41***	−0.28	
	Psychological coping	3.88	1.5	514	4.38	1.4	461	5.47***	−0.35	
	Recognition	4.01	1.4	514	3.94	1.5	461	0.71	0.05	
	Self-esteem	4.75	1.2	514	5.12	1.1	461	4.77***	−0.31	
	Weight	2.82	1.3	514	3.74	1.6	461	10.02***	−0.64	
Elite competiveness										
	Racing competitively after college	3.86	1.1	608	3.71	1.1	538	2.26*	0.13	
	Make it as a pro runner	2.37	1.3	608	1.96	1.1	538	5.67***	0.34	
	National team is a goal	2.66	1.4	607	2.08	1.3	539	7.08***	0.42	
	Love to be a pro athlete	3.84	1.2	607	2.93	1.4	539	11.56***	0.68	
	Take my training to next level	3.99	1.0	605	3.47	1.3	532	7.59***	0.45	
	If I had opportunity, I’d train more	3.46	1.2	607	3.01	1.2	539	6.37***	0.38	
	Train less after college (R)	3.35	1.2	607	3.29	1.2	538	0.95	0.06	
	Elite competitiveness (mean 7 items)	3.35	0.8	603	2.92	0.9	531	8.48***	0.50	
Training volume										
	Mileage per week	69.98	17.4	592	51.69	13.0	505	19.50***	1.21	
	Adjusted mileage per week	69.98	17.4	592	63.06	15.9	505	6.83***	0.42	
	Running sessions per week	8.32	2.2	557	7.30	1.8	487	8.05***	0.51	
	Cross training sessions per week	1.32	1.8	587	1.64	1.9	511	2.86**	−0.17	
	Total sessions per week	9.58	2.8	557	8.85	2.6	487	4.34***	0.27	
Notes.

* p < .05.

** p < .01.

*** p < .001.

Next we explored the seven new items designed to address motivation to train and compete as an elite runner after college. As predicted, these items showed good internal consistency (alpha = .82), supporting the hypothesis that they may constitute a unitary construct that we provisionally call “elite competitiveness.” Supporting its validity, for both men and women, elite competitiveness (mean of seven items) correlated significantly with the SOQ and MOMS measures of competitiveness and goal achievement, our two primary measures of running volume, and 5,000 m best performance quartile (Table 2). In addition, supporting the hypothesis that it may be specifically associated with exceptional performance and training commitment, elite competitiveness was, in most cases, more highly correlated with running volume and running performance than were the SOQ and MOMS measures (Table 2). For men, the correlations were significantly greater (p < 0.05) for elite competitiveness and 5,000 m performance and elite competitiveness and weekly adjusted mileage than were the respective associations for MOMS competitiveness, MOMS goal achievement, or SOQ win orientation. Also, for men the correlation was significantly greater for elite competitiveness and weekly training sessions than for MOMS competitiveness and weekly training sessions or MOMS goal achievement measure and weekly training sessions.

Table 2 Intercorrelations for Competitiveness-related Measures, Training Volume, and Performance.

Measure	1	2	3	4	5	6	7	8	9	10	
1. SOQ: Competitiveness	–	0.61*	0.57*	0.65*	0.54*	0.44*	0.08	0.09	−0.05	0.24*	
2. SOQ: Win Orientation	0.66*	–	0.26*	0.60*	0.35*	0.31*	0.07	0.14*	−0.03	0.28*	
3. SOQ: Goal orientation	0.54*	0.29*	–	0.33*	0.47*	0.29*	0.07	0.05	−0.04	0.14	
4. MOMS: Competition	0.63*	0.55*	0.36*	–	0.70*	0.34*	0.12*	0.11*	0.01	0.29*	
5. MOMS: Goal achievement	0.48*	0.25*	0.46*	0.64*	–	0.35*	0.15*	0.12*	−0.02	0.21*	
6. Elite competiveness	0.37*	0.15*	0.34*	0.23*	0.37*	–	0.13*	0.03	−0.06	0.30*	
7. Adjusted mileage per week	0.15*	0.16*	0.12*	0.05	0.06	0.26*	–	0.46*	0.00	0.46*	
8. Running sessions per week	0.05	0.06	0.04	−0.03	0.02	0.16*	0.57*	–	0.03	0.26*	
9. Cross training sessions per week	0.00	0.08	0.08	0.03	0.00	0.07	−0.07	−0.05	–	−0.23*	
10. 5,000 m Performance quartile	0.22*	0.19*	0.07	0.01	0.02	0.33*	0.55*	0.44*	−0.10	–	
Notes.

Intercorrelations for women are presented above diagonal, and intercorrelations for men are below the diagonal. Correlations for women are based 349 participants, save those involving 5,000 m performance, which are based on 194 participants. Correlations for men are based on 426 participants, save those involving 5,000 m performance, which are based 244 participants.

* p < .05.

As predicted, men scored significantly higher than women in elite competitiveness (d = 0.50), as well as six of the seven items contributing to it (Table 1).

As predicted, men reported significantly greater training volumes (Table 1). This was true for absolute running mileage per week (d = 1.21), adjusted running mileage per week (i.e., increasing women’s mileage by 22%; d = 0.42), and running sessions per week (d = 0.51). Women reported significantly more aerobic cross-training sessions per week than men, but the difference was modest (d = − 0.17). Moreover, runners engage in cross-training only a few times per week but usually run at least once per day (Table 1). Thus, for those runners who reported both running sessions and cross-training sessions, the total number of training sessions per week (sum of running and cross-training) was significantly greater for men (d = 0.27). Table 2 also reveals that greater adjusted mileage and more running sessions were significantly associated with faster running for both men and women but that greater cross-training was only significantly associated with faster running for women.

Sub-population selectivity

Our chief question is whether sex differences in competitiveness and related measures become significantly smaller as a function of sub-population selectivity or running ability. We first addressed this by grouping individuals into sex-specific quartiles based on fastest 5000 m performance and then conducting homogeneity of slopes analyses, testing whether, for each measure, there was a significant interaction between sex and running performance quartile. We focused on the three competitiveness measures, two running volume measures, and MOMS goal orientation and SOQ win orientation. These latter two measures were correlated with competitiveness measures (Table 2) and, across all runners, were endorsed significantly more by men than women.

Figure 1 illustrates, for each performance quartile, the sex difference in each measure as an effect size. Figure 2 shows differences in means for the competitiveness and running volume measures.

Figure 1 Sex differences, as effect sizes, in competitiveness-related measures and training volume as a function of 5,000 m performance quartile.

Figure 2 Sex differences in competitiveness and training volume as a function of 5,000 m performance quartile.

Solid lines indicate women; dashed lines indicate men. Squares indicate means and error bars represent two standard errors of the mean (95% confidence interval). Quartile 1 (Q1) runners are slowest and Quartile 4 (Q4) runners are fastest.

There were significant interactions for MOMS competition (F(1, 534) = 13.3, p = .0002) and MOMS goal achievement (F(1, 534) = 4.4, p = .035). For both MOMS sub-scales, the sex difference decreased among faster runners (Fig. 1). In fact, for runners in the fastest quartile, there was no significant sex difference with either measure (competition: t(129) = 1.23, p = .22, d = 0.22; goal achievement: t(123) = 0.59, p = .56; d = − 0.10).

There were no significant interactions between and sex and running performance in predicting the three SOQ sub-scales (all ps > .50) or elite competitiveness (p = .97; Fig. 1). For SOQ competiveness, SOQ win orientation, and elite competitiveness, the sex difference was significant for all four ability quartiles, with the exception of SOQ win orientation for the second slowest quartile (t(140) = 1.45, p = .15; d = 0.25).

There were significant interactions for adjusted mileage per week (F(1, 607) = 5.9, p = .016) and running sessions per week (F(1, 579) = 10.5, p = .001). In both cases, the sex difference was greater among faster runners (Figs. 1 and 2). The sex difference for both training volume variables was significant (p < .05) for the three fastest quartiles but not for the slowest quartile (running sessions: t(145) = 0.60, p = .55; d = 0.10; adjusted mileage: (t(143) = 1.67, p = .10; d = 0.28).

Our second measure of sub-population selectivity or running ability was the runner’s team’s athletic division, NCAA DI, DII, or DIII. We first confirmed that athletic division was associated with running performance, as assessed by fastest 5,000 m performance quartile: this correlation was significant for women (r(297) = .50, p < .001), and the median best performances for DI, DII, and DIII were 18:04, 18:55, and 19:21. For men also, the correlation was significant (r(331) = .45, p < .001), and the median best performances were 14:54, 15:22, and 15:47.

We next conducted homogeneity of slopes analyses to test whether there were significant interactions between sex and athletic division. As illustrated in Figs. 3 and 4, there was no indication of an interaction (all ps > .35) for any of the competitiveness-related measures. For all of these measures, the sex difference was significant for each athletic division (all ps < .01), with the one exception of MOMS goal achievement for D1 runners (t(440) = 0.97, p = .33; d = 0.09).

Figure 3 Sex differences, as effect sizes, in competitiveness-related measures and training volume as a function of athletic division.

Figure 4 Sex differences in competitiveness and training volume as a function of athletic division.

Solid lines indicate women; dashed lines indicate men. Squares indicate means and error bars represent two standard errors of the mean (95% confidence interval). DIII runners are generally slowest, DII runners are generally intermediate, and DI runners are generally fastest.

Explanations for the sex difference

There are several possible explanations for the sex difference in competitiveness. One candidate is that female runners generally suffer more injuries or illness than male runners, and this could more frequently disrupt their training or diminish their enthusiasm for competing. However, there was no indication of a sex difference for the two items addressing the extent to which injuries or illness compromised training (Table 3).

Table 3 Sex differences in characteristics relevant to potential explanations for the sex difference in competitiveness.

Topic or instrument	Item or sub-scale	Men	Women	t	Cohen’s d	
		M	SD	n	M	SD	n			
Injuries										
	Miss 2+weeks of training due to injury	2.10	2.3	607	2.10	2.3	537	0.04	0.00	
	Had to limit training due to injuries	2.80	1.4	606	2.89	1.4	538	1.08	−0.06	
Children										
	Already have child	0.00	0.0	602	0.00	0.1	530	0.69	−0.04	
	Likely to have child in next 3 years	1.46	0.7	607	1.39	0.7	535	1.74	0.10	
Academics										
	Do my best in school	4.32	0.9	602	4.60	0.6	533	6.09***	−0.37	
	Take education as seriously as running	4.01	1.2	604	4.47	0.9	537	7.45***	−0.45	
	Hours studying	13.67	7.2	607	15.60	7.0	538	4.60***	−0.27	
Notes.

* p < .05.

** p < .01.

*** p < .001.

A second explanation is that women’s lesser competitiveness is due to their having greater obligations or burdens outside of sports. One specific obligation is childcare or the expectation of providing childcare in the near future. Contrary to this explanation, there was no sex difference in having children already or the likelihood of having children in the next three years (Table 3). More importantly, the rates for both were very low. Only one of 530 woman reported having children already, and only 32 of 535 women (6%) responded with a 3 or greater on a 5-point scale (“neither agree nor disagree) to the statement “I am likely to have children in the next 3 years”; only nine (1.6%) responded with a response of 4 or greater (“agree”).

We also addressed the obligations explanation in an indirect but more general way. We did this by including items addressing runners’ commitment to their academic studies. The logic is that if women have more obligations (of various kinds) than men do, then women should also report studying less than men. However, women reported spending significantly more time studying (Table 3), suggesting that female runners generally do not prioritize running as much as male runners do. Supporting this interpretation were sex differences on the two items endorsing commitment to academics; on both items, women scored significantly higher than men (Table 3).

Discussion

An important question about sex differences in preferences and motivation is whether they persist when men and women are drawn from selective sub-populations (Croson & Gneezy, 2009; Ferriman, Lubinski & Benbow, 2009). We addressed this question by investigating competitiveness in intercollegiate distance running. Our main finding is that male runners are, on average, more competitive than female runners, and this sex difference is undiminished among the fastest runners. Strengths of our study include its large sample size, use of two measures of sub-population selectivity, and use of several measures of competitiveness. Before examining the possible cause(s) of this sex difference in competitiveness and its theoretical implications, we consider potential objections and limitations to our study.

Possible objections

One objection to our main conclusion—that the sex difference is undiminished among the fastest runners—is that we found clear evidence that the sex difference disappeared among the fastest quartile of runners for MOMS competition and MOMS goal achievement (Figs. 1 and 2). Although these results are notable, they do not undermine our general conclusion. This is because we did not find any narrowing of the sex difference in SOQ competitiveness, SOQ win orientation, elite competitiveness, or either measure of training volume. Furthermore, Table 2 provides some evidence that MOMS competition and goal achievement, although valid within recreational runners (Masters, Ogles & Jolton, 1993; Masters & Ogles, 1995; Ogles, Masters & Richardson, 1995), may not be in more selective sub-populations. In particular, unlike the other competitiveness-related measures, neither of these MOMS sub-scales were significantly associated with 5,000 m performance in male intercollegiate runners (MOMS competition, r = 0.01; MOMS goal achievement, r = .02). In addition, although men in the present sample reported greater MOMS goal achievement than did women, the sex difference was modest (d = 0.21), and other studies of goal achievement report no sex difference or else greater goal achievement for women (Gill, 1986; Gill & Dzewaltowski, 1988; Gill & Deeter, 1988; Jamshidi et al., 2011).

Another crucial point is that the sex difference in competitiveness did not vary significantly across NCAA athletic divisions for any of the competitiveness-related measures, including the two MOMS measures (Figs. 3 and 4). These results are notable because NCAA athletic division correlated with timed running performance for both women (r = 0.50) and men (r = 0.45), and these tests were based on large samples, at least 975 total runners for each measure (Table 1).

A second objection is that we have found an overall sex difference in competitiveness, but this difference is trivially small and unimportant (Gill, 2000). This objection appears to have little merit with respect to our study. One reason this objection is weak is that the sex differences in competitiveness and training volume we found (ds ∼ 0.50; see Table 1) are comparable in magnitude to those typically found in social psychology as a whole (d = 0.45) and about twice as large as those typically reported for studies of sex differences (d = 0.26; Richard, Bond & Stokes-Zoota, 2003; see also, Hyde, 2005). We also note that the sex differences in competitiveness and training volume varied substantially across performance quartiles (Table 2 and Fig. 2) and athletic divisions (Fig. 4), as expected by definitions of competitiveness (Gill & Deeter, 1988; Masters, Ogles & Jolton, 1993). Nonetheless, Figs. 2 and 4 show that sex differences in competitiveness and training volume were roughly similar in magnitude to differences across performance quartiles and athletic divisions; this pattern indicates that sex explains appreciable variance in competitiveness. A final reason our results should not be dismissed as unimportant is that they appear to explain ecologically relevant behavior. In particular, there is a substantial sex difference in the depth of outstanding American professional distance runners, as about two to three times as many men as women run fast relative to sex-specific standards (Deaner, 2006b; Deaner, 2013). This pattern is what would be expected if, among talented intercollegiate runners, women were less likely than men to be motivated to pursue a professional running career, as indicated in this study (Table 1).

A third objection is that results based on our sample might not generalize to all NCAA distance runners. This objection is valid in the sense that we did not assess the extent to which our sample corresponds to the population. Nonetheless, with respect to our conclusions about a sex difference in competitiveness, this objection appears weak. In the Methods section we showed that, although faster runners tended to be overrepresented as respondents, the greater response of men in our study (28.1%) compared to women (24.3%) was due to slower male runners being significantly more likely than slower female runners to respond. Because running ability is positively correlated with competitiveness (Table 2), this suggests that a more representative sample might yield a slightly larger sex difference in competitiveness than obtained in this study. Furthermore, the men and women who participated did not differ significantly in their responses to several items that might be related to competitiveness, including running experience and the achievement of top performances in conference and national cross-country meets (see Methods).

A fourth potential objection is that our comparisons should have focused on male and female runners whose absolute performances are similar (i.e., relatively fast women, relatively slow men). Figure 2 is pertinent, as women in performance Quartile 1 (defined as having a best 5,000 m performance of 17:54 or faster) can be compared with men in Quartile 4 (defined as having a best 5,000 m performance of 15:58 or slower). These women are significantly higher than these men in elite competitiveness and training volume, and they are similar in SOQ competitiveness and MOMS competitiveness. Thus, one might say, “When we compare men and women of roughly equal absolute ability, the women are at least as competitive as the men, and this result means that, when men and women are drawn from equally selective sub-populations, there is no sex difference in competitiveness”.

This interpretation is provocative, yet it is unsatisfying for several inter-related reasons. One reason is that there is consensus that men have major physiological advantages over women in distance running and most other sports (Sparling, O’Donnell & Snow, 1998; Thibault et al., 2010). Largely due to this, most sports competition, including distance running, is segregated by sex, and this likely has implications for competitiveness. For instance, if a women achieves a performance of 15:30 for the 5000 m run, this indicates she is a contender to become a D1 women’s national champion and could have realistic aspirations to make an Olympic team and to earn substantial prize money as a professional; these considerations must be associated with her elite competitiveness responses (Table 1). By contrast, a man achieving such a performance will be highly unlikely to be a varsity runner (i.e., regularly contributing member) for most D1 men’s teams. Finally, considering absolute performances would mean that nearly all top running performances would be achieved by men. For example, in recent years, none of the women’s best 5,000 m collegiate performances would be included in a (combined or absolute) yearly list of the nation’s top 500 performances (http://perma.cc/WWQ7-NHKY). In sum, we are confident that our decision to make comparisons of men and women of similar ability relative to their own sex is appropriate and would be endorsed by nearly all sports scientists, coaches, and distance runners.

Explanations for the sex difference in competitiveness

There are many candidate explanations for the sex difference in competitiveness in intercollegiate distance runners. One explanation is that female intercollegiate runners have fewer opportunities than male runners to compete and train. Our results suggest that this is unlikely, because women were no more likely than men to have their training disrupted by injuries and illness, almost no women already had children, and few women anticipated having a child in the next three years (Table 3). Moreover, women spent more time studying than men and reported greater commitment to academics than to running (Table 3). These results suggest that, rather than lacking sufficient time for running, female runners are instead more likely than males to prioritize their studies over their running.

A second candidate explanation is that female intercollegiate runners’ lesser competitiveness is due to their having fewer incentives to excel compared to their male counterparts. Much evidence contradicts this idea. Consistent with NCAA data (NCAA Research, 2011), we found that women were more likely than men to receive athletic scholarships. These scholarships can be worth tens of thousands of dollars, and many runners earn them after matriculating at an institution and achieving outstanding performances.

Another kind of incentive is prize money at road races. Intercollegiate athletes are not permitted to accept substantial prize money, but improving one’s performance during college can increase one’s capability to earn prize money after college. To the best of our knowledge, all U.S. road races have identical purses for men and women, and prizes for excelling in international competition, such as the Olympics, are equivalent (Byrnes & Drawbaugh, 2012). It is true that across all international road races, there is estimated to be 15% more prize money available for men than women (Association of Road Racing Statisticians, 2013). However, world-wide, there are roughly eight times as many male as female professional runners who run relatively fast compared to gender-specific world class standards, and the variance among top female performances is substantially greater than among top male performances (Frick, 2011). This lesser depth among professional female distance runners suggests that, all else being equal, it should be easier for women than men to earn prize money. Finally, another important source of income for professional distance runners is sponsorship (e.g., from footwear companies), and a recent survey of U.S. distance runners showed highly similar receipt of sponsorship for men and women (Huntley, 2013).

A third candidate explanation for the sex difference in competitiveness is more general, potentially applying to many kinds of athletes, not only distance runners. It holds that girls and women typically experience less encouragement and opportunities to be competitive (and even participate) in sports. This socialization explanation is popular, and there is much evidence consistent with it (Gill, 2000; Fredricks & Eccles, 2005; Hogshead-Makar & Zimbalist, 2007; Brake, 2010). For example, parents’ perceptions of their children’s sports ability are strongly associated with the children’s sports ability beliefs and their participation (Fredricks & Eccles, 2005). However, the correlational design of studies in this area means that attributions of causality must be viewed cautiously. Similarities between parents’ and children’s sports interest, for instance, might be driven by children’s behavior. Likewise, these studies do not consider the possibility that sports interest (and perhaps competitiveness) may be genetically transmitted, as some studies indicate (e.g., Lykken et al., 1993; Hur, McGue & Iacono, 1996).

Another kind of evidence often cited in support of the socialization view is that the sex difference in sports interest (and perhaps competitiveness) has supposedly declined in the U.S. since the passage of Title IX in 1972. Title IX prohibits sexual discrimination in educational opportunities, including sports, and has resulted in the creation of substantially more equitable opportunities and incentives (e.g., scholarships) for female athletes (Hogshead-Makar & Zimbalist, 2007; Brake, 2010). Although Title IX has been successful in many respects, increasing female participation in organized school sports does not provide unambiguous evidence of increasing female sports interest. This is because there must have been considerable unmet female sports interest prior to Title IX, i.e., girls and women who wished to participate but lacked opportunities. In fact, a recent study showed that outside of organized school settings, the sex difference in sports participation (and presumably interest) remains large in the contemporary U.S., with males participating roughly three times as much as females (Deaner et al., 2012). Furthermore, this sex difference apparently has not decreased since at least the early 2000s (Deaner et al., 2012). Other studies indicate that the sex difference in competitiveness in U.S. distance runners declined somewhat in the 1980s and 1990s but has been stable since (Deaner, 2006b; Deaner, 2013; Deaner & Mitchell, 2011; Deaner, Addona & Mead, 2014).

A fourth explanation is that the sex difference in competitiveness is due, at least in part, to males’ typically greater exposure to prenatal androgens. Although no study in this area has focused on sports competitiveness per se, several kinds of evidence indicate that sex-differentiated childhood activity patterns (e.g., boys greater interest in rough-and-tumble play) are partly due to boys’ greater exposure to prenatal androgens (Berenbaum & Beltz, 2011). These activity patterns, in turn, predict adult sports interest (Giuliano, Popp & Knight, 2000; Cardoso, 2009). Furthermore, females with congenital adrenal hyperplasia, a disease characterized by heightened prenatal androgen exposure, are more likely than unaffected females to show strong interest in stereotypically masculine sports (Berenbaum, 1999; Frisén et al., 2009).

Finally, from a functional, evolutionary perspective, the present study’s results are compatible with the hypothesis that men are more predisposed than women to engage in cultural displays to publically signal their qualities to potential mates, competitors, and allies (Hawkes & Bird, 2002; Lombardo, 2012; Deaner, 2013). It is important to emphasize two things about this hypothesis. First, it is fully compatible with hypotheses addressing proximate causality (e.g., socialization, prenatal androgen exposure). Second, this hypothesis does not hold that men are generally more industrious or capable than women. Instead, the claim is that men are predisposed to channel their efforts into domains where they can “show off’ ’in comparison to others. This hypothesis is supported by several lines of evidence (Deaner, 2013), including that that men generally report stronger competitive orientations, whereas women report stronger work orientations (e.g., Spence & Helmreich, 1983; Gill, 1986). The present results support this hypothesis because intercollegiate distance running, like most other sports, constitutes public competition, and men reported greater motivation to compete. By contrast, academic performance, which women prioritized more, seems less publicly visible.

Explanations for the sex difference persisting among faster runners

Studies of financial decision makers often report that the sex difference in risk taking substantially weakens or disappears when comparing individuals from similar and selective sub-populations (Croson & Gneezy, 2009). The main interpretation is that the generally observed difference in risk taking is mediated by domain-relevant characteristics, such as investment knowledge or wealth. When men and women do not differ in these characteristics—due to selection, training, or endowment—then the sex difference in risk taking disappears (Croson & Gneezy, 2009).

Our study of distance runners yielded a different result, namely that the sex difference in competitiveness was generally undiminished in selective sub-populations. This could be due to one of several differences between the sub-populations of distance runners and financial decision makers. One difference is that women are substantially under-represented among financial decision makers, whereas there are slightly more female than male NCAA intercollegiate distance runners. Thus, if the sport of distance running (like most other sports) was not segregated by sex, and there was selection for competitiveness (but not other characteristics), we would expect that male runners would substantially outnumber female runners, but there would be no sex difference in competitiveness. In other words, one might claim that, although both male and female runners are drawn from selective sub-populations, the male sub-population is actually more selective. Such a pattern might be due to female non-elite populations (e.g., high school runners) being generally less competitive than corresponding male non-elite populations; thus, the threshold to reach the intercollegiate level, in terms of competitiveness and ability, might be lower for women than for men.

A second, related difference is that moderate to high levels of risk taking may be indispensable for financial decision making jobs, but high competitiveness, although helpful in distance running, may not be required. Much more important, perhaps, are a runner’s physiological and biomechanical characteristics. Thus, although male and female intercollegiate distance runners may be identical in many important respects, such as their likelihood of possessing genes associated with high maximal oxygen uptake (Tucker & Collins, 2012), they may differ in other characteristics, including competitiveness. This interpretation is supported by the fact that, although the competitiveness scores for our distance runners (see Table 1) are higher than those of recreational athletes (Gill, 1988; see also Deaner et al., 2011), they are unexceptional compared to other amateur athletes (Hellandsig, 1998; Jamshidi et al., 2011), and they are substantially lower than those of professional tennis players (Houston, Carter & Smither, 1997).

Our finding that the sex difference in competitiveness remains undiminished among the faster runners is broadly consistent with previous studies of distance running. One study reported that, in the U.S., two to four times as many males as females ran fast relative to sex-specific world class standards, and this difference was nearly as large among professional distance runners as it was among high school and collegiate runners (Deaner, 2006b). In the study showing a sex difference in marathon pacing, the sex difference increased significantly among slower runners, yet even among the fastest runners (e.g., sub-3 h finishers), men were roughly three times as likely as women to experience marked slowing (Deaner et al., 2015).

Our results are also consistent with research on lifestyle preferences among adults in their mid-30s identified as profoundly gifted as children or as top math or science graduate students about ten years earlier (Ferriman, Lubinski & Benbow, 2009). Although men and women in these selective sub-populations were matched in their scientific abilities, their adult preferences differed. Men generally focused on their careers, including gaining recognition for their professional achievements, whereas women reported holding a more communal orientation, balancing careers with friendships and family. These sex differences emerged as individuals transitioned from their training (e.g., graduate school in mid-20s) to their careers and had or anticipated having children (Lubinski et al., 2001; Ferriman, Lubinski & Benbow, 2009).

Although our distance running results echo those of the top science ability study (Ferriman, Lubinski & Benbow, 2009), they differ in three notable ways. First, our distance running results chiefly addressed competitiveness, not lifestyle preferences. Second, the sex difference we documented cannot be as readily attributed to women anticipating that the conflict between professional and family goals will be more acute for them than for men. This is because most collegiate runners likely enjoy their greatest opportunities for earning recognition (e.g., winning individual and team championships) and financial rewards (i.e., scholarships) during their collegiate careers, a time when almost none have or anticipate having children (Table 3). Even the very small number of runners with the talent and desire to succeed as professionals generally reach their earnings peaks by their late 20s (Huntley, 2013). This is because elite endurance performance declines with age beginning at roughly age 30 (Joyner, 1993). Because the median age for bearing a first child for college-educated women in the contemporary U.S. is 30 (Hymowitz, 2013), these points suggest that most collegiate female runners aspiring to excel as professionals would only be modestly deterred by the possibility that running would interfere with motherhood.

Third, our results are notable because the sex difference in competitiveness shown here occurs in a sex-segregated domain, whereas in science and most other settings, men and women jointly compete. This is notable because, in behavioral economics experiments, women are substantially less competitive than men in mixed-sex settings (when potential competitors are known to be both men and women) but, in single-sex settings, the sex difference in competitiveness weakens and sometimes disappears (Niederle & Vesterlund, 2011). Thus, in many ecologically relevant (i.e., “real world”) settings, where potential competitors include both men and women, the sex difference in competitiveness might be greater than reported here for distance running.

Conclusions

Sex differences in preferences and motivations are well known and believed to have important implications (Wilson & Daly, 1985; Browne, 2002; Hakim, 2006; Croson & Gneezy, 2009; Ceci et al., 2014). Although the causes of these differences remain unresolved, one popular hypothesis is that if men and women are drawn from selective populations, so that their opportunities and abilities do not differ, then the sex differences in preferences and motivations will shrink or disappear (Croson & Gneezy, 2009). The present study provides one of the strongest tests yet of this hypothesis, and the results did not support it. A crucial question, however, is whether the patterns documented here among intercollegiate U.S. distance runners will generalize to other populations, including those in other societies and other domains of selectivity (e.g., tennis: Houston, Carter & Smither, 1997).

If the main finding of the present study—that sex differences in preferences and motivations remain substantial in selective sub-populations—does turn out to be generalizable, then this may have implications for policy decisions. First, if a policy goal is to facilitate women’s achievement, then policies should consider women’s preferences. In academia, for example, successful policies may include creating part-time tenure-track positions, stopping the tenure clock, and providing subsidized child care (Mason, Wolfinger & Goulden, 2013; Ceci et al., 2014). Some have even suggested that parental leave policies may be most effective if they are reserved exclusively for women (Rhoads & Rhoads, 2012). A second implication is that policy makers may need to reconsider efforts to equalize the number of men and women in some domains where women are under-represented. The present study, along with others (e.g., Hakim, 2006; Ferriman, Lubinski & Benbow, 2009; Lippa, 2010; Ceci et al., 2014), suggests that the under-representation of women may sometimes reflect sex differences in preferences that cannot be easily ascribed to sex differences in incentives and opportunities. Although it is often assumed that these preferences are suboptimal for women, the limited data available suggest otherwise (Pinker, 2009; Ceci & Williams, 2009). For example, in the top science ability study (Ferriman, Lubinski & Benbow, 2009), women with children were less satisfied with their careers than women without children, yet they reported greater life satisfaction. Life satisfaction was not addressed in the present study, yet we suspect that female distance runners giving priority to studying over running similarly reflects the pursuit of their own long-term interests.

Supplemental Information

Supplemental Information 1 College distance runners data

Click here for additional data file.

Supplemental Information 2 Internal consistency and factor structure of MOMS and SOQ sub-scales

Click here for additional data file.

We thank Renah Farhan for assistance with data entry, Michael Joyner for critical encouragement, and Ben Winegard for comments on an earlier draft.

Additional Information and Declarations

Competing Interests

Author Contributions

Human Ethics

1 Several modes of indirect competition have also been identified, and these may be employed more by women than men. These include competing discreetly by disavowing that one is competing (Benenson, 2014), socially excluding competitors (Benenson et al., 2013), and gossiping or spreading rumors about competitors (Archer & Coyne, 2005). There are several published reviews that address possible cause(s), including evolutionary causes, of sex differences in the use of competitive modes (Campbell, 1999; Benenson, 2013; Benenson, 2014).

2 We were unable find results for the 2009 DIII championships, but this is unlikely to affect the reported patterns. This is because for DI and DII results we found that very few runners who were on a roster in the spring of 2013 and finished in the top 50 in 2009 failed to finish in the top 50 at least once in 2010, 2011, or 2012. In particular, there were only three men and two women from DI and DII combined who achieved this distinction, and none of these runners were recruited to participate in our study.

3 Even in the slower quartiles, most intercollegiate runners are of high ability compared to recreational runners. For example, fewer than 1% of performances in 5,000 m road races in the U.S. are fast enough to qualify for the second slowest quartile (i.e., 19:29 or faster for women, 15:57 or faster for men; see (Deaner, 2006a).

The authors declare there are no competing interests.

Robert O. Deaner conceived and designed the experiments, performed the experiments, analyzed the data, wrote the paper, prepared figures and/or tables.

Aaron Lowen performed the experiments, analyzed the data, reviewed drafts of the paper.

William Rogers analyzed the data, reviewed drafts of the paper.

Eric Saksa performed the experiments, reviewed drafts of the paper.

The following information was supplied relating to ethical approvals (i.e., approving body and any reference numbers):

The Chair of the Human Research Review Committee at Grand Valley State University reviewed the study protocol [427545-1] and certified it as approved and exempt from full committee review on February 28, 2013.

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
