# Peer review of "Does the sex difference in competitiveness decrease in selective sub-populations? A test with intercollegiate distance runners"

_PeerJ, doi:10.7717/peerj.884_

## Round 0.1 · original submission · Minor Revisions

Dear Rob and colleagues,

We have received three reviews for your manuscript. All of these are positive and raise a number of useful comments and questions that could be addressed to further enhance the contribution and impact of your work. I would like to thank the reviewers for their timely reviews but also for detailing their comments and suggestions clearly. I won’t revisit them in my own comments as they are well outlined by the reviewers.

I hope the minor revisions feel a suitable use of your time.

Reviewer 1 ·

Basic reporting

No comments

Experimental design

No comments

Validity of the findings

1. Why are there no sex differences in competitiveness in tennis? What is the difference between running and tennis?
2. Why are there no sex differences in this study using the MOMS measure of competitiveness for the fastest runners, but there are sex differences using the SOQ? How do the two measures differ?
3. Why did the authors choose to examine sex differences in running, as they state that men are 10% faster than women? What would happen if sex differences were compared for runners of the same speed?
4. The authors report that Benbow and colleagues find that high-performing male math students develop their careers more than their female counterparts. But is this competitiveness?
5. Lines 673-676 don’t make sense. Much research suggests that females behave more competitively with males than with other females. Of course, if females are competing for males, then this is different. Regardless, I don’t understand the sentence that follows, just before the Conclusion.
6. Finally, speculation about the proximate reasons males compete more than females seems unfounded given the lack of data. It would be valuable however if the authors explained the theoretical reason for their interest. From an evolutionary perspective, it seems that females may be more afraid than males to compete overtly, as retaliation is more probable and injury is more detrimental to women’s reproductive success than men’s (Anne Campbell, 1999). It may be that males stand to gain more than females from defeating rivals due to the greater benefits of multiple mates for males (Trivers, 1972). The contribution of the manuscript would be greater if the importance of sex differences in competition for understanding human nature were highlighted. Then, if the authors provided some theoretical explanation for why elite performers should or should not exhibit the same sex differences, the results could be more easily assimilated.

·

Basic reporting

No comments

Experimental design

No comments

Validity of the findings

No comments

Additional comments

This is a well-conducted study on an interesting topic: Sex differences in competitiveness in selected populations. The paper describes its methods and results very thoroughly, and the literature review is also thorough. I commend the authors for reporting their results in terms of the effect size d.

I have just one suggestion for some issues I didn’t see discussed – that levels of competitiveness are likely to be very high in this population regardless of gender. Making it to the NCAA, and particularly making it into the top quartile of performance, presumably requires a great deal of competitive drive. Thus this is a view of sex differences at the top of the distribution. In most cases, looking at the extremes of a distribution creates larger differences. For example, the sex difference in aggression is about d = .50 on average, but five to six times as many men are incarcerated for violent crimes. The sex difference on the math SAT isn’t that large at the average, but multiplies for those who score several SDs above the mean. In general, if you look at two bell curves a half a standard deviation apart and examine the area under the curve two or three standard deviations out, there’s a bigger difference at the top and bottom.

This argument holds more strongly for this population than others studied (e.g., gifted children) as there likely is a link between being included in the population (college runners) and the variable of interest (competitiveness). Intelligence (as in the gifted children study) might not be strongly linked to motivation (plenty of smart but lazy people), but just getting to the status of elite running (the population here) is likely connected to competitiveness.

There’s also the possibility the gender interacts with competitiveness in selecting people into this population. If women are generally less competitive, perhaps those who are more competitive will have an easier time reaching a high level of performance (if, for example, they faced less competition early on, because fewer women were competitive by nature). That would mean that women who were even moderately competitive would be more likely to make it into running, thus preserving the sex difference even at the high level.

I don’t think these possibilities are detrimental to the paper, but it might be beneficial to include them in the introduction or discussion to round out the presentation of ideas.

·

Basic reporting

See below

Experimental design

See below

Validity of the findings

See below

Additional comments

This is a nicely-designed, reasoned, and analyzed study. None of my comments below are meant to imply otherwise. I think it will make a valuable contribution—not only to the literature on competitive running but to sex differences among elite groups more generally.

Who are the 80 NCAA institutions that sponsored women’s but not men’s cross country teams--mainly women’s colleges?

p. 5: I am confused by this statement: “Participants reported an average of 2.2 previous seasons of collegiate cross country and 5.5 previous seasons of total collegiate distance running, including cross-country, indoor track and field, and outdoor track and field.” How can they have on average a total of 5.5 seasons of collegiate running? What am I missing?

Why were previous seasons of total collegiate distance running not included?

The authors’ evolutionary argument makes sense, along with more proximal factors they mention. It is surprising and important that male runners are typically more competitive than female runners, and this sex gap is not narrowed among the fastest runners.

But evolutionary claims will likely be met with objection, with some grasping at the data in Figures 1 and 2 showing that sex difference disappeared among the fastest quartile of runners for MOMS competition and goal achievement measures. Perhaps some attention to this in addition to the present argument will preclude such argument.

There is, as these authors note, an interesting contribution that is secondary to their running findings. Namely, among an elite group of men and women the gap has not closed or even really narrowed, despite resources. While this is not central to this paper it might serve as the basis for an essay or editorial in another venue. There are data on sex differences in participation in elite math competitions such as the Putnam and Olympiad contests. Andreescu et al (2008) have some data in the Notices of the American Mathematical Society, Vol 55, No. 10. 1248-1255. Ellison and Swanson (2009) have an NBER report that argues that the bulk of highly competitive girls in math are in a handful of high schools. In other words, they argue that if the culture of high schools changed to be like the ones where all of the girls hail from who enter math contests, there would be many more girls in the elite category. Along these lines, there is the counterargument—which I do not find compelling—that sex differences among elite groups are driven primarily by cultural forces. None of this work has been with elite athletes, of course, but the finding that sex differences in math and science are narrowed as a function of a nation’s gender equality index is invoked to dismiss evolutionary arguments. BTW, I think Ellison and his colleagues have contrary findings on this point.

In sum, this is a nicely-designed study that has implications beyond the domain of elite runners. Due to its large sample size, coupled with the use of multiple measures of sub-population selectivity and measures of competitiveness, I think it will be cited broadly once the word gets out.

---

## Round 0.2 · accepted · Accept

Dear Rob and colleagues,

I am please to let you know that I have received feedback from two referees (Stephen Ceci declined to review the revision, indicating that he was happy with the manuscript and his suggestions were entirely at the authors' discretion), and they are happy to accept the work as is. One reviewer notes an interesting future study that follows from the work.

Reviewer 1 ·

Basic reporting

No comments

Experimental design

No comments

Validity of the findings

No comments

Additional comments

The authors have addressed the issues raised. I continue to wonder however whether sports in which females are as or more talented than males, such as gymnastics or figure skating, would reduce sex differences in competitiveness. I suppose that is a question for another manuscript.

·

Basic reporting

No comments

Experimental design

No comments

Validity of the findings

No comments

Additional comments

I think the authors have done a great job responding to the reviews, and I look forward to seeing the paper published.